# Emerging Roles of Cells and Molecules of Innate Immunity in Alzheimer’s Disease

**DOI:** 10.3390/ijms241511922

**Published:** 2023-07-25

**Authors:** Bartolo Tamburini, Giusto Davide Badami, Marco Pio La Manna, Mojtaba Shekarkar Azgomi, Nadia Caccamo, Francesco Dieli

**Affiliations:** 1Department of Biomedicine, Neuroscience and Advanced Diagnosis (BIND), University of Palermo, 90127 Palermo, Italy; bartolo.tamburini@unipa.it (B.T.); giustodavide.badami@unipa.it (G.D.B.); marcopio.lamanna@unipa.it (M.P.L.M.); mojtaba.shekarkarazgomi@unipa.it (M.S.A.); francesco.dieli@unipa.it (F.D.); 2Central Laboratory of Advanced Diagnosis and Biomedical Research (CLADIBIOR), AOUP Paolo Giaccone, 90127 Palermo, Italy

**Keywords:** microglia, Alzheimer’s disease, neuroinflammation

## Abstract

The inflammatory response that marks Alzheimer’s disease (neuroinflammation) is considered a double-edged sword. Microglia have been shown to play a protective role at the beginning of the disease. Still, persistent harmful stimuli further activate microglia, inducing an exacerbating inflammatory process which impairs β-amyloid peptide clearance capability and leads to neurotoxicity and neurodegeneration. Moreover, microglia also appear to be closely involved in the spread of tau pathology. Soluble TREM2 also represents a crucial player in the neuroinflammatory processes. Elevated levels of TREM2 in cerebrospinal fluid have been associated with increased amyloid plaque burden, neurodegeneration, and cognitive decline in individuals with Alzheimer’s disease. Understanding the intricate relationship between innate immunity and Alzheimer’s disease will be a promising strategy for future advancements in diagnosis and new therapeutic interventions targeting innate immunity, by modulating its activity. Still, additional and more robust studies are needed to translate these findings into effective treatments. In this review, we focus on the role of cells (microglia, astrocytes, and oligodendrocytes) and molecules (TREM2, tau, and β-amyloid) of the innate immune system in the pathogenesis of Alzheimer’s disease and their possible exploitation as disease biomarkers and targets of therapeutical approaches.

## 1. Introduction

Neurodegenerative Alzheimer’s disease (AD) is the most common type of dementia, accounting for around 60–80% of all dementia cases [1]. The prevalence of AD rises with age and affects approximately 10 to 30 people over the age of 65 years [2]. This disease has many risk factors, including age, gender, cardiovascular diseases (CVD), cerebrovascular diseases, and metabolic diseases [3]. The pathological features of AD are characterized by the presence of plaques of β-amyloid (βA) peptide and neurofibrillary tangles (NFTs) of the phosphorylated protein tau, leading to progressive memory and cognitive impairment [4]. Recent developments indicate that systemic inflammation affects the central nervous system (CNS) to cause neurodegeneration and cognitive decline, highlighting the close relationship between the immune system and the CNS. In the CNS, the most important immune cells are the microglia, which on one side can switch towards an activated state to remove βA through phagocytosis but which on the other can cause neuronal damage due to the release of proinflammatory molecules [5]. In Alzheimer’s disease, this inflammatory reaction, known as neuroinflammation, represents, therefore, a double-edged sword. Recent evidence suggests an important role of inflammation in the pathophysiology of AD as demonstrated by the identification of the pro-inflammatory cytokines IL-1α, IL-1β, IL-6, TNF-α, granulocyte-macrophage colony-stimulating factor (GM-CSF), and IFN-α in the AD brain produced by neurons or microglia, refs. [6,7,8,9,10] endothelial cells and inflammatory cells recruited from the circulation, and by the blood–brain barrier (BBB) when biochemical or mechanical damage occurs [11,12].

Other proinflammatory elements are represented by the complement system, acute phase proteins, and immune cell subsets [13,14].

Several clinical and preclinical studies were performed to understand the key role in AD of neuroinflammation, as well as histopathological and neuroimaging analyses or proteomic analysis and other biomarker signatures in cerebrospinal fluid (CSF) and in blood. Microglial activation has been observed at the pre-plaque stage in animal models of AD [15] and in a study of human neuroimaging where increased microglial activation in people with mild cognitive impairment (MCI) in the absence of amyloid tracer uptake was observed [16,17,18,19,20,21].

In addition to innate immunity and inflammation, the adaptive immune system may also play a role as suggested by the detection of B and T lymphocytes in the post-mortem AD brain and CSF of MCI and AD patients [22], as well as by the increased frequency of T helper subsets in the pathogenesis of AD [23,24,25,26,27,28,29,30].

Furthermore, the contribution of B cells to AD pathogenesis is still unclear and controversial, even if memory B cells have been found to increase in the CSF of MCI patients and correlate positively with βA deposition in the brain [27]. In conclusion, many types of innate and adaptive immune cells may contribute to the pathogenesis of AD [31,32,33,34,35].

In conclusion, many types of immune cells, involving both innate and adaptive immune systems may contribute to the pathogenesis of AD [17,18,19,20,21]. Here, we describe the role of resident innate immune cells such as microglia, astrocytes, and oligodendrocytes, while the role of the other cells is described elsewhere.

This review aims to dissect the contribution of the cells mentioned above and molecules (tau, βA, and TREM-2) of the innate immune compartment to the pathogenesis and progression of AD and their potential as disease biomarkers, and the design of tools toward these targets.

## 2. The Role of Microglia in Alzheimer’s Disease

Several complicated alterations in innate and adaptive immunity occur in neurodegenerative disorders [36]. The main innate immune cells in the brain that are involved in inflammatory responses are microglia and astrocytes [37], which are strongly implicated in alterations of molecular pathways in AD. Microglial cells represent approximately 10% of brain tissue and play an active role in complex neurodevelopment such as neurogenesis and synaptic pruning [38]. This cellular population interacts with and responds to immunological or neuronal stimuli in the microenvironment, displaying a key role in homeostasis and immune surveillance of the brain. Microglial cells sense changes in the microenvironment, removing pathological agents [37,39,40], producing cytokines and chemokines, and recruiting other immunocompetent cells to protect against noxious stimuli.

In AD, the loss of homeostasis due to tissue injury induces various microglial changes, including changes in surface phenotype or cell morphology and proliferative responses, ref. [41] highlighting the complex role that this cellular subset displays in regulating the right balance of its beneficial or detrimental role.

Microglial cells express molecules belonging to the innate compartment named pattern recognition receptors (PRRs) that can recognize several pathogen-associated molecular patterns (PAMPs) or damage-associate molecular patterns (DAMPs). Various species of βA, including the neurotoxic oligomer βA, bind PRR expressed on microglial cells. Still, the subsequent accumulation of βA determines their activation and causes the neuroinflammation observed in AD [42,43].

Microglial cells are both beneficial and detrimental in Alzheimer’s disease [44]. At early stages of the disease, microglial activation has been linked to neuroprotective effects by suppressing harmful stimuli represented by βA hyperproduction, on releasing neurotrophic factors and clearing and phagocyting debris or dead cells [45]. However, persistent harmful stimuli across the disease continuum, such as DAMPs, further induce the microglial activation by initiating an auto-perpetual pro-inflammatory process that leads to βA clearance ability [39,46]. This determines a chronic activation with subsequent release of inflammatory cytokines, which leads to neurodegeneration and neurotoxicity.

Recent research has emphasized the critical role of microglia in the pathogenesis of AD, especially the processing and dissemination of tau protein, a defining feature of the illness. Extracellular tau clumps can be engulfed and degraded by microglia. These cells engage through phagocytosis to engulf aberrant proteins or cellular debris that they find using their surveillance capabilities. Following phagocytosis, the engulfed material is destroyed with the aid of lysosomes, which include enzymes for dissolving proteins. This process might aid in halting the development of tau disease. When tau is ingested by microglia, they occasionally bundle it into tiny vesicles called exosomes, and release it back into the extracellular space. When nearby neurons pick up these tau-containing exosomes, tau pathology spreads across the entire brain [44,47].

Before microglia cells can respond to a stimulus, they must first be attracted to the stimulus site, which is commonly mediated by the interaction of chemokines and cytokines with their receptors, both of which are expressed by microglial cells. The formation of βA plaques in the brain is one of the hallmarks of AD disease. These plaques are composed of misfolded and aggregated βA peptides that are thought to cause neuronal damage and cognitive impairment. Microglial cells play an essential role in the phagocytosis of βA plaques, and misfolded βA peptides can attract them. Indeed, the abundance of chemokine receptors expressed by microglia and chemokines generated by βA-stimulated cells are suggestive of their essential role in microglia accumulation in AD. In this context, in vitro studies show that βA-activated microglia can secrete various chemokines, including CCL4, CXCL2, CCL3, CXCL8, and CCL5 [48].

In AD, neuroinflammation is a prominent feature and is believed to contribute to disease progression [49,50]. Several inflammatory proteins and genetic factors associated with innate immunity in AD, including Apolipoprotein E (APOE) and Triggering Receptor Expressed On Myeloid Cells 2 (TREM2), as well as Macrophage Migration Inhibitory Factor (MIF), soluble forms of TREM1, and TREM2 [51,52,53]. These proteins exhibit distinct expression profiles at different stages of AD, suggesting their potential as biomarkers for disease progression.

Among these proteins, MIF is increased in both MCI and AD stages [54]. Studies have shown that MIF levels correlate with amyloid plaques in the frontal cortex, indicating its involvement in βA pathology [53,55]. Interestingly, MIF has also been proposed as a neuroprotectant, preventing microglia-mediated neuronal loss induced by βA. The release of MIF by microglia and neurons under various stressors suggests its role as an alarmin, recruiting and activating microglia to respond to pathological insults.

In the last decades, the need to deeply analyze in vivo the role of innate immunity cell populations and pathogenic mechanisms involved in the onset and progression of AD made it necessary to adopt animal models. The most common are mouse models which have some limitations [56]. The mouse brain cannot naturally develop amyloid plaques. Consequently, transgenic animal models expressing human mutations linked to precocious-onset AD are helpful in investigating AD-like pathology [57]. In many of these transgenic AD models, a combination of human genes carrying the Swedish double mutation (KM670/671NL) for amyloid precursor protein (APP) and mutations of the Presenilin 1 (PSEN1) gene, such as the APPPS1 mouse strain, is used.

This mouse model displays in aging the formation of βA plaques which are surrounded by activated microglia [58] in a closely similar way to that commonly observed in humans [59,60].

The microglia, which cluster around amyloid deposits, display dysfunctional behavior and cannot effectively clear amyloid plaques [61]. The lowering in βA clearance might be attributed to gene expression changes. Interestingly, the researchers observed a significant decrease in the surface expression of microglial receptors binding βA, such as the CD36 molecule, a scavenger receptor A (SRA), and the receptor for advanced-glycation end products (RAGE). Concurrently, the microglia showed a marked increase in the expression of the pro-inflammatory cytokines IL1-β and TNF-α [61].

The study findings indicate that in the APPswe/PSEN1dE9 mouse model of AD, microglia exhibit impaired amyloid plaque clearance, likely attributed to alterations in gene expression, accompanied by a shift towards a pro-inflammatory cytokine profile.

In the transgenic experimental mouse model expressing five familial mutations observed in early-onset AD families (5×FAD), the human IgG1 monoclonal antibody aducanumab-targeting fibril and oligomer of βA, administered by focused ultrasound (FUS), triggered the activation of phagocytic microglia and increased the number of astrocytes associated with amyloid plaques in the hippocampus of these mice [62].

Moreover, applying combined treatment to 5×FAD mice resulted in notable changes in the hippocampus, as revealed by RNA sequencing. Specifically, the enrichment of four canonical pathways was observed: phagosome formation, neuroinflammation signaling, CREB signaling, and reelin signaling. These alterations in pathways seem to play a crucial role in the treatment’s effectiveness.

Notably, the combined treatment reduced amyloid deposits, addressing a key hallmark of AD [62].

An elegant study carried out on a novel *APP* knock-in mouse model (*App*^SAA^) of AD with three mutations (Austrian, Swedish, and Arctic) in the mouse *APP* gene, showed significant alterations regarding lipid metabolism and disease-associated transcriptomic signature in the microglia containing a high amount of intracellular βA, consisting of vascular amyloid deposits, an accumulation of parenchymal amyloid plaques, alteration of astroglial and microglial functions, and an increase in CSF markers of neurodegeneration. The fibrillar βA content in microglia is associated with foam cell phenotypes, lysosomal dysfunction, and lipid dysregulation [63].

In a mouse model of C57BL/6 humanized *APP* mutant knock-in homozygote (App^NL-G-F^) treated with Adeno-associated virus expressing P301L tau mutant, it was assessed that the expression of tau mutant in the medial entorhinal cortex (MEC) determined tau propagation to the granule cell layer of the hippocampal dentate gyrus; this process was exacerbated in App^NL-G-F^ mice in comparison with WT control mice. This result suggests that neurodegenerative microglia (MGnD) hypersecrete p-tau^+^ extracellular vesicles (EVs) despite compacting βA plaques and clearing NP tau; this result correlates with the plaque deposition and exacerbation of tau propagation in App^NL-G-F^ mice [64]. The administration of PLX5622, a highly selective brain penetrant colony-stimulating factor 1 receptor (CSF1R) inhibitor, eliminate almost all microglia in mouse brains and significantly reduced propagation of p-tau in both WT and App^NL-G-F^ mice. However, it increased plaque burden and plaque-associated p-tau^+^ dystrophic neurites.

To study the microglial role in tau fibril pathologies, other mouse models have been developed, such as the htauP301S and the PS19, both with the human microtubule-associated protein tau (MAPT) gene mutated and inducing frontotemporal dementia. In these models, microglia are significantly activated and clustered, surrounding tau-positive neurons [65]. In PS19 mice, the microglial activation was observed before the formation of a tangle [66], similar to what was observed in AD patients, where altered microglial function preceded the formation of tauopathy [67].

Some mouse strains, such as 3xTg-AD, try to replicate βA and tau diseases. This strain bears the human transgenes for APP, PS1, and tau with pathogenic mutations.

In this model, the βA plaques and NFT formation are related to the increased number of F4/80 microglia/macrophages found in the entorhinal cortex starting from 6 months of age [68], and by the number of Iba1-positive microglia increasing later in the hippocampus region [69].

Another mouse AD model able to develop βA and tau pathology is the APPsweDI/NOS2^−/−^, previously designed to explore the role of nitric oxide (NO) [70].

This mouse bears the APP Iowa, Dutch, and Swedish mutations and is knock-out for the nitric oxide synthase (NOS)2 gene encoding the inducible (i) NOS enzyme, which is closely associated with oxidative stress and neuroinflammation. These mice exhibit elevated levels of tau hyperphosphorylation and aggregation, accompanied by an increase in microglial activation, an important memory impairment after the neuronal loss. Interestingly, it is worth noting that after few months of life, these mice develop tau pathology even in the absence of the human tau transgene, indicating a significant correlation between inflammation and the formation of tau tangles [70].

Finally, another mouse model of β-amyloidosis shows increased C1q levels and enhanced synaptic localization of C1q even before plaques have developed [5]. It is well known that C1q binding to βA determines the classical complement cascade activation. This finding has been observed in human AD brain tissue using the immunohistochemistry technique. In fact, complement proteins (C1q, C3, and C4) have been found, especially in plaques and, to a lesser extent, in neurofibrillary tangles and dystrophic neurites (C5b–C9). The adult brain expresses C1q, and as people age, their protein levels sharply increase, particularly in the hippocampus [44,71].

Neutralizing antibodies against C1q or genetic C1q knockouts prevents the synapse loss seen in amyloid-bearing mice or brought on by βA injections [5].

Thus, identifying the role of disease-specific microglial phenotypes and functions in the progression of AD may be relevant in preclinical mouse models to the design of immunotherapy that stimulates or alleviates inflammation as the disease progresses.

## 3. The Role of Astrocytes in Alzheimer’s Disease

In the damaged CNS, astrocytes are thought to be critical regulators of innate and adaptive immune responses [72,73]. Astrocytes, once considered as mere supportive cells, are now recognized as critical players in the pathogenesis of AD, due to their proinflammatory effects [74], since, similar to microglia, they release proinflammatory cytokines such as IL-1β, TNF-α, and IL-6 [75]. An increased amount of proinflammatory cytokines has been detected in the BBB of AD patients compared to healthy subjects, indicating chronic neuroinflammation that can contribute directly to the neurotoxicity observed in AD. Moreover, astrocytes trigger the microglial activation [76] and participate in the clearance of βA plaques through phagocytosis. However, in AD, astrocytes become less efficient in clearing βA, leading to its accumulation. This impaired clearance is partly attributed to the downregulation of crucial βA transporters, such as Low-density lipoprotein receptor-related protein 1 (LRP1), in astrocytes.

In addition to βA plaques, neurofibrillary tangles—a hyperphosphorylated tau protein—are typically of AD [77,78]. Emerging evidence suggests that astrocytes actively participate to the propagation of tau pathology. Astrocytes take up and spread pathological tau through tunneling nanotubes, contributing to the propagation of tau aggregates throughout the brain [79,80]. Astrocytes also disrupt tight junction proteins and release matrix metalloproteinases (MMPs) to facilitate inflammatory cells infiltration and disruption of the BBB integrity [81]. Astrocyte involvement in neuroinflammation, synaptic dysfunction, tau pathology, and BBB integrity dysfunction highlights their complex role in disease progression and as innate immunity regulators. Targeting these cells holds promise as a therapeutic strategy. Modulating astrocyte-mediated neuroinflammation, enhancing βA clearance, and promoting synaptic function are potential intervention strategies. Additionally, strategies aimed at improving astrocytic energy metabolism and reducing tau propagation could be explored.

Ugolini et al. found that reactive astrocyte numbers increased in AD mouse models, notably those surrounding βA plaques, and that the size and number of main astrocyte branches decreased. They contended that various brain regions may exhibit diverse astrocyte responses to the same stimuli, leading to differential neuronal survival status. This may help to explain why astrocyte atrophy and astrogliosis do not manifest in all areas of the brain simultaneously in AD [82].

Reactive astrogliosis is known to increase with age in the cerebral cortex of 5×FAD mice compared to WT mice. On the other hand, the transcription factor E2F4 can prevent or delays the cognitive impairment observed in 5×FAD mice [83,84]. A mouse model of 5×FAD with a supplemental mutation that silences the neural expression of the gene coding for E2F4 and expresses enhanced green fluorescent protein (EGFP) as a marker of astrocyte activation showed the role of activated astrocytes in 1-year-old mice which were more significant in mice with a silent E24F gene compared to control mice [85].

In a recent study, Muraleedharan et al., utilizing the 5×FAD mouse model of AD, uncovered the significant involvement of PKC-eta (PKCη), an astrocyte-specific stress-activated and anti-apoptotic kinase, in reactive astrocytes. They illustrated the pronounced enrichment of PKCη staining within cortical astrocytes, particularly in proximity to βA peptide plaques, in a disease-dependent manner [86].

They demonstrated that inhibiting the kinase activity of PKCη in 5×FAD astrocyte cultures leads to a marked increase in the levels of secreted IL-6. Intriguingly, a similar phenomenon is observed in wild-type astrocytes when stimulated by inflammatory cytokines [86].

Another recent research has highlighted that some astrocytes exhibit aberrant and abundant production of the inhibitory gliotransmitter GABA, achieved through monoamine oxidase-B (Maob) activity. Additionally, these astrocytes release GABA abnormally via the bestrophin 1 channel [87].

In mouse models of AD, this released GABA has a detrimental effect on granule cells in the dentate gyrus. It significantly reduces the spike probability of these cells by acting on presynaptic GABA receptors, thereby impacting synaptic plasticity and impairing learning and memory.

When GABA production or release from reactive astrocytes is suppressed, a complete restoration of impaired spike probability, synaptic plasticity, and learning and memory is observed in the mice.

Moreover, a significant upregulation of astrocytic GABA and MAOB in the postmortem brains of individuals with AD is observed, further reinforcing the relevance of this study [87].

## 4. The Role of Oligodendrocytes in Alzheimer’s Disease

Oligodendrocytes are a type of glial cell found in the CNS that play a crucial role in the formation and maintenance of myelin. This fatty substance wraps around nerve fibers and provides insulation. Myelin is essential for the proper functioning of the nervous system, including efficient transmission of electrical signals between neurons [88].

In AD, several pathological changes occur in the brain. While the primary hallmark of AD is the accumulation of βA plaques and tau tangles, recent research has also implicated the involvement of oligodendrocytes in the disease process, because myelin disruption can lead to the degeneration and death of neurons, impairing their ability to transmit signals effectively. This can result in cognitive impairment, memory loss, and other neurological deficits associated with AD [89,90].

Several human and animal studies have investigated oligodendrocyte changes in AD; these abnormalities include decreased numbers of oligodendrocytes and alterations in the structure and function of myelin. The loss and dysfunction of oligodendrocytes can disrupt the integrity of myelin and lead to impaired communication between neurons. A noteworthy phenomenon was observed among 6–8-month-old APPPS1 mice.

The findings revealed a marked increase in the population of OPCs (oligodendrocyte precursor cells) within this specific group. However, a contrasting trend emerged when examining postmortem tissues of individuals affected by AD. The number of Olig2^+^ cells, a marker for mature oligodendrocytes, was significantly decreased in these AD patients [91].

An additional study revealed an increased abundance of remyelinating oligodendrocytes expressing MAP-2 in the regions neighboring periventricular white matter lesions. Furthermore, it observed a greater presence of OPCs expressing PDGFR-α within the white matter lesions [92].

A PS1 knock-in mouse model has shown that oligodendrocytes are highly susceptible to glutamate and βA-induced damage, leading to increased vulnerability and eventual cell death. Furthermore, these cells exhibit a notable impairment in their ability to regulate calcium levels, exacerbating their susceptibility to neurotoxic insults [89].

Another study has revealed findings regarding myelin basic protein (MBP) and the population of myelinating oligodendrocytes in 6-month-old triple transgenic mice (3xTg-AD).

The researchers observed a decrease in the number of myelinating oligodendrocytes and in MBP levels. Surprisingly, they observed the same number of immature oligodendrocytes, with an increase in mature non-myelinating cells, suggesting that myelinating oligodendrocytes are particularly vulnerable to oxidative stress. This vulnerability stems from their higher metabolic demand and their increased iron and lipid content. This sensitivity to oxidative stress could potentially be contributing to the observed reduction in MBP and myelinating oligodendrocytes in triple transgenic mice [93].

A cell culture study conducted on rat oligodendrocytes proved that oxidative stress induced by βA can lead to the death and dysfunction of these cells. The study further revealed that βA toxicity for oligodendrocytes may involve mitochondrial DNA damage and subsequent activation of NF-kB and AP-1 as alternative mechanisms [94].

Furthermore, research suggests that the interaction between oligodendrocytes and other pathological features of AD, such as βA plaques and tau tangles, may contribute to disease progression.

## 5. Tau and βA Proteins

The typical profile that emerges in people with dementia mainly concerns abnormalities in the activation of tau, βA and microglia (Figure 1). Intraneuronal hyperphosphorylated tau aggregation results in a variety of neurodegenerative conditions known as tauopathies, of which AD, which appears to be the leading cause of dementia, is a secondary tauopathy.

Tau is a protein expressed predominantly in the brain and is associated with microtubules. Its weight is equal to 55kDa and in the human brain it is present in six isoforms encoded by a single gene called MAPT located on chromosome 17q21 by alternative mRNA splicing. Tau promotes neuronal microtubule assembly by binding to tubulin via repeats on its binding domain. However, tau can dissociate from microtubules in people with AD, aggravating the disease [95]. The diminished propensity of tau to bind to microtubules due to its aberrant hyperphosphorylation is the main cause of microtubule instability in various tauopathies [96]. Tau is also involved in axonal development, elongation, transport, regulating iron homeostasis, myelination, nuclear architecture, and neurogenesis [97,98]. All processes involving tau protein in neurodegeneration concern its phosphorylation, particularly in the proline-rich site located in the C-terminal region. This post-translational modification is widespread in tau. It is favored by various kinases, in particular by proline-directed kinases such as glycogen synthase kinase β (GSK3β), cyclin-dependent kinase 5 (CDK5) and microtubule affinity-regulating kinase (MARK) [97,99]. Hyperphosphorylated tau protein is also the major constituent of abnormal neurofilament tau (NFT), which in various neurodegenerative stages can twist around each other, creating a coupled helical filament (PHF). This PHF can accumulate in the neural perikarial cytoplasm, axons, and dendrites, causing deregulation and loss of cytoskeletal microtubules and tubulin-associated proteins [100]. Tau that has been inappropriately phosphorylated builds up and loses its affinity for microtubules, leading to axonal transport dysfunction and overall neurotransmission problems. Due to increased oxidative stress, tau and βA continue to accumulate at synapse sites, resulting in the loss of dendritic spines, presynaptic terminals, and axonal dystrophy [99,101]. The prevention of tau aggregation can be an exciting strategy to slow AD progression. Small compounds and other disease-modifying therapies (DMT) can be utilized to prevent the first stage of tau aggregation, hence reducing the accumulation of the protein [100]; specifically, several studies have focused on preventing tau hyperphosphorylation and buildup by inhibiting particular kinases such as GSK3 with drugs such as tideglusib and saracatinib. The latter is still in phase II trials and has shown remarkable results in enhancing memory in transgenic mice [102]. AD has been associated with many tau-related biomarkers [103]; for example, exosomes containing phosphorylated tau have been found in the cerebrospinal fluid (CSF) of early AD patients and the amount of tau protein in CSF can be correlated with the severity of cognitive impairment [104,105]. This finding makes the CSF levels of released tau and βA [1,2,3,4,5,6,7,8,9,10,11,12,13,14,15,16,17,18,19,20,21,22,23,24,25,26,27,28,29,30,31,32,33,34,35,36,37,38,39,40,41,42] promising biomarkers for early AD detection [106].

For instance, CSF p-tau 217 (tau phosphorylated at threonine 217) is specific for AD and can be used to aid in the differential diagnosis of other illnesses. Dramatically increased CSF levels of p-tau 217 are helpful in both the early and late stages of AD diagnosis [97,107]. According to a recent study, the cellular absorption of tau, which occurs in distinct phases, is necessary to spread tauopathies [108]. Starting with the dimerization of two modified tau monomers, intermediate soluble oligomers with different higher order conformations and phosphorylation levels are created. Toxic tau oligomers serve as “tau seeds” that attract common monomers to create new aggregates, resulting in fresh neuro fibrillary tangles (NFTs) that impair cognitive performance. Currently accepted theory predicts that these tau seeds subsequently go to the neuronal somatodendritic compartment and form NFTs, resulting in cognitive impairment. Although there is evidence that tau trimers are the minimal unit of spontaneous cellular uptake and intracellular fibrillary structure development in vivo, the folding potency of monomers may be far more critical in starting the early nucleation process of tau aggregation [108,109].

Chronic neuroinflammation forces microglial transformation into disease-associated microglia (DAM). According to Keren-Shaulet et al., DAM is characterized by overexpression of specific genes such as TREM2, TYRO, APOE, and protein tyrosine kinase binding protein (TYROBP), and downregulation of genes involved in homeostasis [110]. Additionally, microglia are essential for avoiding amyloid plaque burden and assisting in defense against AD pathogenesis [111]. They encapsulate the βA plaques to form a barrier that stops their spread and lowers βA’s capacity to harm neurons by phagocytosis. The transcription factor IRF8 is required for the paradoxical ability of microglia to disseminate βA across the brain, as demonstrated by a recent study [112]. The transcription factor IRF8 promotes microglial migration to the tissue damage site and enhances the spread of βA plaques by microglia. Microglial cells surround and engulf existing βA plaques to avoid the spread of βA plaques; on the other hand, they support the formation of new βA plaques through the transport and release of ingested βA seeds into different brain regions [113]. The buildup of βA-peptide in synapses activates microglia, causing them to release TNF-α, IL-1, IL-6, reactive oxygen species (ROS), nitric oxide (NO), and other immunomodulatory compounds into the extracellular environment, accelerating neurodegeneration. Tau, in addition to βA, boosts the inflammatory response in microglia by activation of the inflammasome (NLRP3), IL-1, and NF-κB dependent pathways. NLRP3 activation, induced by both βA and tau, initiates the assembly of its components, increasing caspase-1 cleavage and furthering the release of IL-1 and IL-18, which causes cell death through the process of “pyroptosis” [9,114]. Thus, microglial cells and the various immune system activation stages are closely related. A key to prevent or at least reduce neuronal dysfunction may lie in a better knowledge of the processes governing these interactions and, most importantly, the extracellular levels of tau.

## 6. TREM1 and TREM2

TREM2 is a protein expressed on the surface of microglia and other immune cells that plays a role in innate immunity in the brain. Recent studies have implicated TREM2 in AD pathogenesis (Figure 2). TREMs are transmembrane glycoproteins associated with DAP12 for signaling and function [115,116], an expanding family of receptors including the activating and inhibitory isoforms encoded by MHC-associated clusters. The TREM1 isoform is involved in inflammation, while the other isoform, TREM2, regulates the functions and development of osteoclasts and microglia.

The importance of TREM2 is related also to the context of non-immune cells because the presence of genetic defects in its expression results in brain and bone diseases. Several in vivo studies have confirmed the role of TREM1 as an amplifier of inflammation [117,118]: it is expressed by neutrophils and monocytes [115] and TREM1 expression amplifies TLR-initiated inflammatory responses to microbial challenges by inducing production of IL-8, myeloperoxidase, TNF and IL-1β [115,117,119]. Recent data are consistent with the theory that TREM1 promotes monocyte chemokine production (IL-8, CCL2, and CCL3) that attract effector Th1 cells to the site of inflammation [120].

TREM2 is expressed by DCs generated from immature monocytes and primarily regulates the function of other myeloid cells, such as phagocytosis, cytokine release, and modulation of microglial morphology, in contrast to TREM1, which primarily regulates granulocyte and monocyte/macrophage responses [116]. Dysregulation of TREM2 signaling has been associated with impaired microglial response and compromised clearance of toxic protein aggregates in AD. The brain and microglia both express TREM2 and DAP12 [121,122].

There is growing evidence that TREM2 is a crucial regulator that enables the transition of microglia from a homeostatic to a disease-associated state [110,123] and several studies on mouse models have already demonstrated its role [122,124,125]. However, whether TREM2, or his surrogates, could be used as a possible non-invasive biomarker of AD, remains elusive.

The impact of soluble TREM2 (sTREM2), the endogenous function of sTREM2 itself, and the downstream TREM2 receptor signaling, are not entirely clarified; in particular, the role of sTREM2, once released from the membrane, is not well understood. After shedding, sTREM2 could play an agonistic function by binding a particular TREM2 ligand on different cells, or it could have a role as a decoy molecule competing with the transmembrane ligand binding TREM2 receptor.

What is clear is that under circumstances of enhanced microglial activation, the increased TREM2 protein expression and sTREM2 levels in CSF are higher [126,127], such as in inflammatory neurologic conditions. sTREM2 levels have been extensively evaluated in the CSF and blood in the context of several neurological disorders. Still, these observations did not demonstrate any connection with the levels of sTREM2 in peripheral blood or with markers of BBB integrity, suggesting its intrathecal generation [126,128].

Another study in a cohort of 127 subjects with autosomal dominant AD found that CSF sTREM2 levels were abnormally raised 5 years before the predicted start of symptoms and rose with disease progression in AD, with the highest levels occurring in the early stages of symptoms and indicating changes in microglial activation status in response to neuronal loss [129]. After initial alterations in βA and tau biomarkers, changes in sTREM2 occurred, indicating that microglial activation is a byproduct of these pathogenic occurrences [129]. Another study examined two cohorts (AD/MCI/control cohort and an AD/control cohort) evaluating the use of CSF sTREM2 as a biomarker in AD. In this study, sTREM2 levels in CSF correlated with aging in controls, and with the neurodegenerative markers CSF T-tau and P-tau, but not with Aβ42 [130].

The link between sTREM2 levels in the CSF and other AD biomarkers is the subject of another investigation in which CSF sTREM2 levels were strongly correlated with CSF T-tau and P-tau but not with βA42 in 101 individuals with either amyloid-positive or amyloid-negative status.

A recent study investigated the potential link between baseline levels of CSF sTREM2 and the rate of clinical deterioration and cognitive decline in AD [52]. They examined CSF samples from 385 elderly individuals, including cognitively normal controls, those with mild cognitive impairment (MCI), and subjects diagnosed with AD dementia.

The study findings revealed an intriguing association: higher concentrations of sTREM2 in CSF at baseline were linked to a more gradual decline in memory and cognition among subjects with AD, as identified by evidence of CSF Aβ42.

Regarding clinical progression, in participants who were positive for CSF Aβ42 and CSF p-tau181, a larger ratio of CSF sTREM2 to CSF p-tau181 concentrations indicated a slower transition from cognitively normal to symptomatic phases or from MCI to AD dementia, suggesting that sTREM2 is related to attenuated cognitive and clinical deterioration.

Moreover, in a mouse model of tau pathology, an insightful study by Katsumoto et al. [131] revealed that a mutation with loss of function in TREM2 intensifies chronic inflammation, thereby promoting neurodegeneration in the damaged brain of hTau mice.

Notably, the presence of abnormal BBB leakage in the white matter underscores the indispensable role of TREM2-positive cells in safeguarding against the infiltration of peripheral macrophages.

TREM2 represents a crucial player in the neuroinflammatory processes underlying AD. Its involvement in microglial function, βA metabolism, tau pathology, and genetic interactions highlight the multifaceted role of TREM2 in AD pathogenesis. Understanding the intricate mechanisms of TREM2 signaling and its crosstalk with other pathways may open novel therapeutic approaches or biomarkers targeting neuroinflammation in AD. Elevated levels of sTREM2 in CSF have been associated with increased amyloid plaque burden, neurodegeneration, and cognitive decline in individuals with AD. sTREM2 levels in CSF have shown correlations with other established biomarkers for AD, such as tau and βA. In summary, sTREM2 has shown promise as a biomarker for AD and its measurement in CSF has provided valuable insights into the inflammatory processes associated with AD.

However, it is important to note that sTREM2 levels alone may not be sufficient for an accurate diagnosis of AD. They are typically used with other clinical assessments and biomarkers to increase diagnostic accuracy. Further research is needed to fully understand the role of sTREM2 and establish its clinical utility as a standalone biomarker for AD.

## 7. Illuminating the Immune Landscape of Alzheimer’s Disease: Insights from Bulk and Single-Cell RNA Sequencing

Combining bulk and Single-cell RNA sequencing (scRNA-seq) could shed light on the immunological environment of AD.

Researchers could have access to the molecular alterations occurring in certain brain regions that impact the pathophysiology of AD by studying gene expression patterns through bulk RNA sequencing.

According to their bulk RNA sequencing study, Guennewig et al. have evaluated differential gene expression into two different areas of the post-mortem brains from AD patients: the mildly affected primary visual cortex (VIC) and the moderately affected precuneus (PREC). The authors found that the two regions had similar transcriptomic signatures even if the precuneus showed significant abnormal expression of tau load. Both regions were characterized by upregulation of immune-related genes such as those encoding TREM 2 and MS46A and milder changes in insulin/IGF1 signaling [132].

From this comparison, the greater tau load in the AD-PREC is related to a more remarkable gene expression profile change than that displayed in the AD-VIC. scRNA-seq plays an essential role in understanding AD by providing a high-resolution view of AD pathology with transcriptional alterations associated with the cellular level. scRNA-seq allows the identification of disease-associated cellular subpopulations within a heterogeneous cell population. By analyzing individual cells, specific subpopulations that contribute to AD pathology can be identified, providing a deeper understanding of the disease mechanisms and potential therapeutic targets [133,134].

Recent studies have shed light on the transcriptional heterogeneity of microglia in the human brain. The different microglial polarization status types has been defined as classical activation (M1), alternative activation (M2a), type II alternative activation (M2b), and acquired deactivation (M2c) [135]. In RNA samples extracted from AD brains, the different M1, M2a, M2b, and M2c phenotypes have been detected by co-expression of surface markers in samples [136,137].

The study conducted by Young et al. provides a comprehensive transcriptional map of human microglia at a population scale and elucidates on the decisive role of these cells in CNS development and disease [138]. The study identified distinct subpopulations of microglia and revealed the transcriptional profiles associated with different microglial states. This study did not specifically focus on the different microglial phenotypes, but there has been some debate about how microglial polarization could be related to neurodegenerative conditions such as AD [139].

Studies suggest that microglia can transition between different activation states and exhibit a continuum of intermediate phenotypes between M1 and M2. In neurodegenerative diseases, including AD, microglia-mediated neuroinflammation plays a complex role, with harmful and helpful effects. Balancing the M1/M2 polarization of microglia has shown promise as a therapeutic approach for neurodegenerative diseases, including AD [139].

While more specific research may be needed to identify the distinct microglia subsets in the context of AD through single-cell sequencing, the concept of microglial polarization and its modulation as a potential treatment avenue remains an area of active investigation.

Another innate resident immune cell in the CNS, the astrocyte, plays essential roles in maintaining brain homeostasis and supporting neuronal function. Astrocytes can undergo phenotypic changes in response to various pathological conditions, including AD. The categorization of astrocyte phenotypes into A1 and A2 states has been proposed to describe their reactive responses in neurodegenerative diseases. Although the specific classification of astrocyte proliferation into A1 and A2 states in AD is not well studied, the concept of astrocyte heterogeneity and its role in AD pathology has been discussed by Monterey et al. as the potential basis for the effectiveness of scRNA-seq technologies in identifying specific biomarkers associated with reactive astrocytes, which provides valuable insights into astrocyte contributions and their phenotypic changes in the disease [140].

A recent study using scRNA-seq identified both shared and distinct gene programs induced in astrocytes in AD, suggesting their involvement in the disease pathogenesis [141].

These findings highlight the importance of astrocytes in AD and suggest that their dysfunction may contribute to the pathophysiology of the disease. Further research is needed to fully understand the specific roles of astrocytes and the molecular mechanisms underlying their involvement in AD.

In summary, integrating bulk and scRNA-seq techniques could shed light on the immune landscape of AD. These novel approaches have revealed early changes in disease pathology, identified key molecular players such as βA and tau, and provided insights into the immune response and DNA damage response associated with AD.

These findings contribute to our understanding of the disease and may pave the way for the development of targeted therapies and personalized treatment strategies.

## 8. Possible Treatment Strategies

Research studies have investigated microglia and the progression of AD. One therapeutic approach is microglial depletion, even if the consequences could be critical in the maintenance of brain health, highlighting the need for further investigation [139,142,143].

An intriguing approach as a therapeutic strategy is to target the pro-inflammatory pathways, rather than depleting the microglia.

The inflammasome plays a pivotal role in microglial activation through the induction of the release of pro-inflammatory cytokines, representing an encouraging target for a treatment strategy for AD [144,145]. Some targets of the inflammasome pathway, such as purinergic receptors in NLRP3 and Caspase-1, are promising [146].

Inhibition of purinergic receptors, especially the P2 × 7R subtype found on microglia, in both in vivo and in vitro studies, has been involved in the development neurodegenerative disorders. The use of an antagonist of this receptor offers a therapeutic benefit to understand better its role in neuroinflammation and other neurodegenerative conditions [147,148,149,150,151].

Several molecules have displayed the potential capability to modulate the NLRP3/caspase-1 inflammasome pathway in AD. Several compounds have been found to inhibit the activation of the inflammasome and the production of proinflammatory mediators in the BV2 microglial cell line [152,153] or in experimental mouse models [154,155,156].

Caspase-1 is another target molecule in AD; in fact, caspase-1 inhibition has a neuroprotective effect, further highlighting a potential target molecule as a therapeutic strategy for AD [157]. A natural compound, Bacopa, displays inhibitory effects on several enzymes such as MMP-3, caspase-1, and caspase-3, which are involved in neuroinflammation and neurodegeneration [158], suggesting its potential as a therapeutic agent for AD. In AD mouse models, VX-765, a selective caspase-1 inhibitor molecule, also has promising effects [159].

Another molecule involved in AD progression is TREM2.

The cumulative evidence suggests that TREM2 plays a beneficial role in the CNS, which could be enhanced to prevent or slow down the progression of sporadic AD [160,161,162,163].

Currently, some promising strategies for modulating TREM2 activity have emerged. Two monoclonal antibodies that enhance TREM2 signaling through crosslinking have been discovered [160,161,163,164].

According to this antibody-enhancing strategy, in an experimental mouse model (5×FAD), the development of monoclonal antibody Ab-T1, which is reactive against the extracellular domain of TREM2, attenuated neuroinflammation and improved cognitive function. This monoclonal antibody targets membrane-bound and soluble TREM2, inducing microglial activation. It has also revealed high cross-reactivity with human TREM2. Ab-T1 binds and activates TREM2, enhancing the uptake of labeled βA by macrophages and microglia and promoting TNF-α production. Of note, Ab-T1 enhances the ability of microglia to phagocytose labeled apoptotic neurons, considered as cell debris present around the βA plaque environment [165].

Among the monoclonal antibodies activating TREM2, several studies in transgenic AD mouse models such as 5×FAD mice have evaluated the biological role of AL002a/c. This humanized monoclonal IgG1 antibody binds to TREM2 and activates its signaling pathway [161,162]. The antibody AL002c did not enhance the clustering of microglia around amyloid plaques, while AL002a did. Moreover, AL002a significantly reduced the total βA plaque amount and decreased fibrillar βA within the hippocampus. Furthermore, both AL002a and AL002c reduced the plasma concentration of the neurofilament light chain, which is a neurodegeneration marker [163].

The third approach is focused on preventing the shedding of the TREM2 extracellular domain by ADAM family proteases [162,166].

In this approach, the selective inhibition of TREM2 shedding is preferred over the inhibition of ADAM10, due to the cleavage of another split break of substrates in the brain [162,167].

The monoclonal antibody 4D9 holds promise, as it targets an epitope situated 12 amino acids N-terminal to the TREM2 protein. This strategic binding effectively prevents the shedding of soluble TREM2, making it a highly therapeutic candidate.

In conducted in vitro biochemical assays, 4D9 has displayed its remarkable ability to inhibit the split break of the TREM2 stalk peptide. Consequently, this inhibition leads to the activation of the essential DAP12/phosphorylation/Syk signaling pathway, which plays a pivotal role in mediating downstream signaling events crucial for the functioning of TREM2. These findings suggest that 4D9 holds the key to modulating TREM2-related pathways and provides a valuable avenue for therapeutic interventions in conditions associated with dysregulated TREM2 signaling [160,161,163].

An exciting field of study concerns Nanoparticle-based (NPs) anti-aging treatments that could ameliorate AD more significantly by playing a role as anti-inflammatory tools. Lipid-based nanoparticles (NPs) possess favorable characteristics such as low toxicity and immunogenicity. However, when it comes to delivering drugs across the BBB and targeting specific areas related to AD, liposomes face challenges in facilitating direct transport via passive means [168]. In another study conducted by Meng et al., an approach was undertaken to enhance brain-targeted delivery of curcumin. The researchers developed a novel nanostructured lipid carrier (NLC) known as Cur-loaded Lf-mNLC, which incorporated lactoferrin (Lf) as a modification. The results of this study exhibited remarkable advancements in mitigating pathological neuronal damage in rats [169]. Nanogels (NGs) represent an advanced category of nano-sized three-dimensional polymers that offer great temperature and optical controllability. These polymeric structures were achieved through various chemical and/or physical crosslinking techniques, resulting in enhanced drug release capabilities [170]. Extensive research findings continue to support the crucial role of insulin in protection from AD [171]. In light of this, a groundbreaking advancement has emerged in the form of an NGs system that enables targeted insulin delivery to the brain [172].

## 9. Conclusions

Chronic and self-maintaining inflammation is the main driving force that characterizes the pathogenesis of AD. Acute inflammation, as previously stated, is a critical characteristic of innate immunity that initiates clearance and repair in infected or damaged tissues.

In the neuroinflammatory response, microglia and astrocytes play a crucial role at different stages of brain disease. Given the emerging role of innate immunity in AD, there is growing interest in the development of new therapies that target the innate immune compartment. Various approaches are being explored to modulate microglial function and neuroinflammation. Preclinical studies and phase I/II clinical trials aimed at targeting TREM2 [161] and NLRP3 [173] have displayed promising results. Therapeutic interventions also targeting innate immunity hold promise for slowing disease progression, but additional research is needed to translate these findings into effective treatments. Understanding the intricate interplay between innate immunity and AD will pave the way for future advancements in diagnosis and therapy.

However, though encouraging, these therapeutic studies are still preliminary and need further improvements to have a deeper comprehension of the close relationship of the above-described cells and molecules and their contributions to the onset and progression of AD.

## Figures and Tables

**Figure 1 ijms-24-11922-f001:**
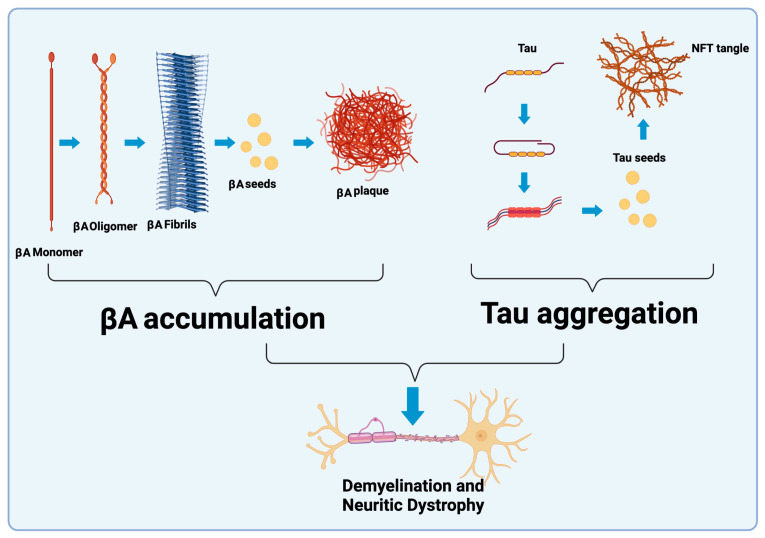
Schematic representation of AD pathology characterized by extracellular βA aggregates, NFT, and subsequent demyelination and neuritic dystrophy.

**Figure 2 ijms-24-11922-f002:**
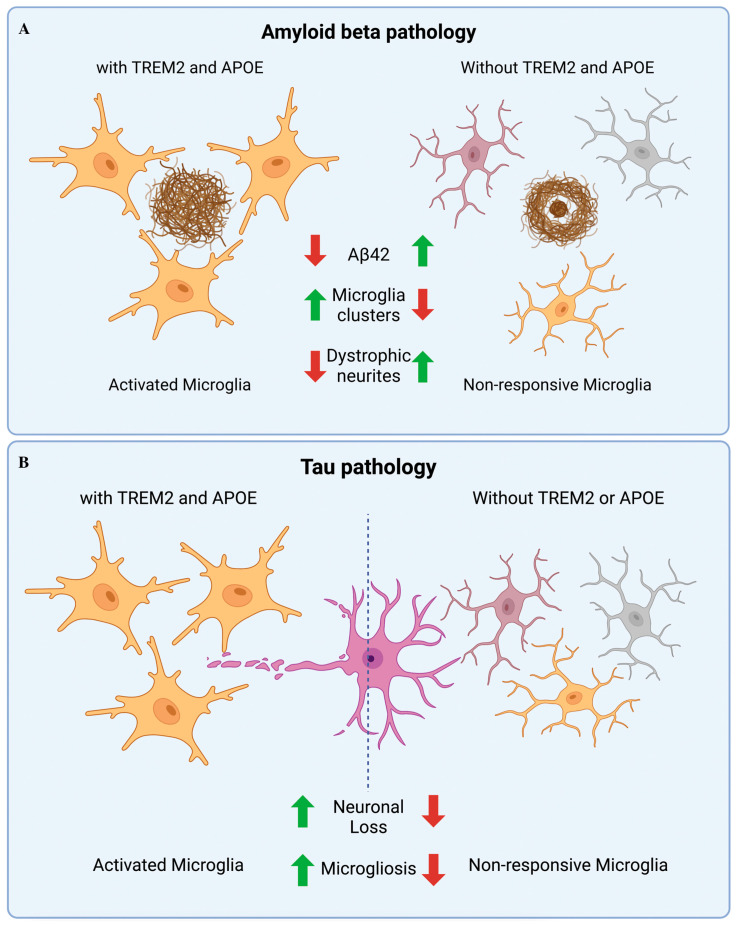
Microglial functions in Alzheimer’s disease (AD): (**A**) TREM2 and APOE work together in βA-bearing disease to reduce neuronal damage by driving microglial clustering. DAM is defective in the absence of TREM2 or APOE, and filamentous plaques have numerous amyloid plaques and dystrophic neurites. (**B**) In tauopathy TREM2 or APOE deficiency controls microgliosis and prevents neuronal loss. A green arrows mean upregulation; a red arrow means downregulation; the dotted line separates activation of microglial cells from non-activation status.

## Data Availability

Not applicable.

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
