# Peer review of "Emerging Roles of Cells and Molecules of Innate Immunity in Alzheimer’s Disease"

_ijms, 2023, doi:10.3390/ijms241511922_

Round 1

Reviewer 1 Report

This review manuscript describes the role of inflammatory molecules in the context of AD. This manuscript is well written and the information provided in the manuscript fits the overall goal of the paper. I have a few comments for this review paper.

1) Sections 2 and 3: Microglia and astrocytes and more so microglia are known to be major players in the development of neuroinflammation in various neurodegenerative models. However, the authors have given very little context for each of these cells. I think this paragraph should contain more material with regards to different models used, microglial effects found in these models of neurodegeneration, etc.

2) I think the authors could add another section of treatment strategies of targeting the cell types discussed for alleviating neuroinflammation rather than just making a comment in the conclusion section.

A few minor English editing may be required for this manuscript. Otherwise it is fine.

Author Response

Reviewer 1: Comments and Suggestions for Authors

This review manuscript describes the role of inflammatory molecules in the context of AD. This manuscript is well written and the information provided in the manuscript fits the overall goal of the paper. I have a few comments for this review paper.

  • Sections 2 and 3: Microglia and astrocytes and more so microglia are known to be major players in the development of neuroinflammation in various neurodegenerative models. However, the authors have given very little context for each of these cells. I think this paragraph should contain more material with regards to different models used, microglial effects found in these models of neurodegeneration, etc.

    Thank you for your suggestion, we added a new part to expand the section of microglia (line 109-118 and 147-233) astrocytes (line 264-299) and their functions in neurodegeneration models.

  • I think the authors could add another section of treatment strategies of targeting the cell types discussed for alleviating neuroinflammation rather than just making a comment in the conclusion section.

    Thank you. We included this point in paragraph 8 and cited more current developments (line 599-683)

Reviewer 2 Report

Overall the information presented represents valuable information regarding the roles of cells and molecules of innate immunity in Alzheimer’s disease.  I have a few suggestions for the manuscript

1.      Oligodendrocytes are myelinating cells in the CNS, oligodendrocytes are also reported in Alzheimer’s disease. Include the role of oligodendrocytes together with microglia and astrocytes.

2.      Please mention why other cell types in the immune system are not included in the review (in short).

3.      In section 2 The Role of microglia in Alzheimer's Disease, including interaction and processing of Tau protein.

4.      Include recent developments in bulk and single-cell RNA seq in microglial (M1, M2a, M2b, M2c, M2d) and astrocyte proliferation (A1 and A2) in AD context.

Author Response

Reviewer 2: Comments and Suggestions for Authors

Overall the information presented represents valuable information regarding the roles of cells and molecules of innate immunity in Alzheimer’s disease.  I have a few suggestions for the manuscript

  • Oligodendrocytes are myelinating cells in the CNS, oligodendrocytes are also reported in Alzheimer’s disease. Include the role of oligodendrocytes together with microglia and astrocytes.

We appreciate your feedback and have added a new paragraph that addresses it (line 302-347).

  • Please mention why other cell types in the immune system are not included in the review (in short).

We paid particular attention to some cells and mediators related to the special issue’s subject.

  • In section 2 The Role of microglia in Alzheimer's Disease, including interaction and processing of Tau protein.

We updated the interaction of microglia with Tau protein in paragraph 2 (line 109-118)

  • Include recent developments in bulk and single-cell RNA seq in microglial (M1, M2a, M2b, M2c, M2d) and astrocyte proliferation (A1 and A2) in AD context.

We updated the text about bulk and single-cell RNA sequencing using the most recent information (line 530-597).
